# ActiveMark: on Watermarking of Visual Foundation Models via Massive Activations

## Abstract

Being trained on large and vast datasets, visual foundation models (VFMs) can be fine-tuned for diverse downstream tasks, achieving remarkable performance and efficiency in various computer vision applications. The high computation cost of data collection and training motivates the owners of some VFMs to distribute them alongside the license to protect their intellectual property rights. However, a dishonest user of the protected model's copy may illegally redistribute it, for example, to make a profit. As a consequence, the development of reliable ownership verification tools is of great importance today, since such methods can be used to differentiate between a redistributed copy of the protected model and an independent model. In this paper, we propose an approach to ownership verification of visual foundation models by fine-tuning a small set of expressive layers of a VFM along with a small encoder-decoder network to embed digital watermarks into an internal representation of a hold-out set of input images. Importantly, the watermarks embedded remain detectable in the functional copies of the protected model, obtained, for example, by fine-tuning the VFM for a particular downstream task. Theoretically and experimentally, we demonstrate that the proposed method yields a low probability of false detection of a non-watermarked model and a low probability of false misdetection of a watermarked model.

## 1 Introduction

Today, foundation models are deployed in different fields, for example, in natural language processing (Radford et al., 2019; Brown et al., 2020), computer vision (Ramesh et al., 2022), and biology (Ma & Wang, 2023). Their impressive performance in a wide range of downstream tasks comes at a price of high training cost on large and vast datasets. Consequently, since their training is costly both in terms of time and money, the models become valuable assets of their owners: the user's access to foundation models is mainly organized via subscription to a service where the model is deployed or via purchasing the license to use a specific instance of the model. Unfortunately, some users may violate the terms of use (for example, by integrating their instances of models into other services to make a profit). Hence, it is reasonable that the models' owners are willing to defend their intellectual property from unauthorized usage by third parties.

One of the prominent approaches to protecting the intellectual property rights (IPRs) of models is watermarking (Uchida et al., 2017; Guo & Potkonjak, 2018; Li et al., 2022), the set of methods that embed specific information into a model by modifying its parameters. In watermarking, ownership verification is performed by checking for the presence of this information in a model. An alternative set of methods for IPR protection is based on fingerprinting, which typically does not alter the original model (Pautov et al., 2024; Quan et al., 2023; He et al., 2019). Instead, these methods generate a unique identifier, or fingerprint, for the model; ownership verification is then conducted by comparing the fingerprint of the original model with that of the suspicious model.

This work introduces a method for watermarking visual foundation models (VFMs) by embedding digital watermarks into the hidden representations of a specific set of input images. To select a suitable hidden representation for the watermark, the concept of massive activations is utilized (Sun et al., 2024): certain blocks within a VFM can produce high-magnitude activations in response to various inputs; these activations often dominate the ones in subsequent layers. In this paper, the block where these outputs first occur is referred to as an expressive block. Within the framework,

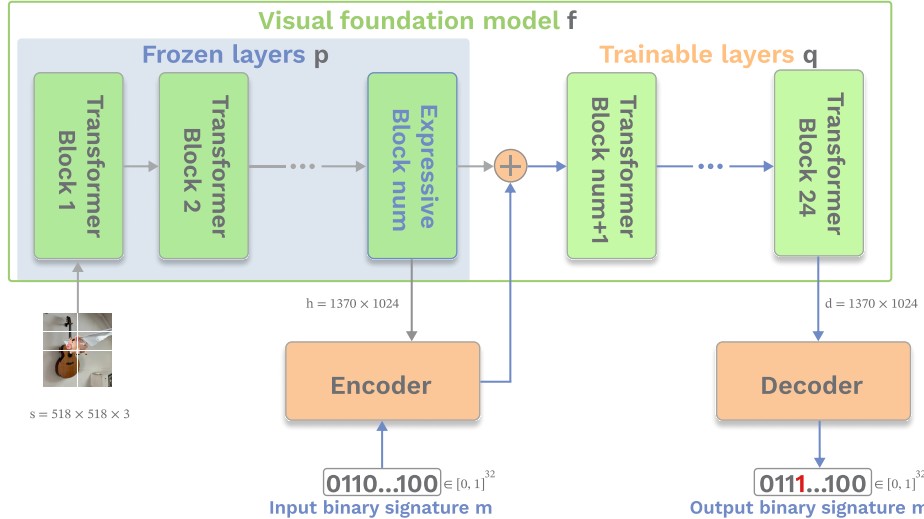

Figure 1: Schematic illustration of the proposed method. To embed the watermark, we pass an image $x$ to obtain its latent representation $p(x)$. Then, the first channel of $p(x)$, namely, $p_1(x)$ is concatenated with the watermark $m$ and passed to the encoder that produces the vector $e(\text{concat}(p_1(x), m))$. Later, the vector $e$ replaces the first channel of the original internal image representation, $p(x)$, and the updated vector $\tilde{p}(x)$ is passed to the latter part of VFM. To extract the watermark, we use a decoding network $d$ that maps an output of the VFM in the form $u = q(\tilde{p}(x))$ to the binary message $m'$, where $m'_i = \mathbb{1}(d(u)_i \geq 1/2)$. Both encoder and decoder are represented by two fully connected layers.

we experimentally verify that embedding a watermark into the representation of the expressive block allows us to protect the ownership of VFMs fine-tuned for different practical tasks, such as image classification and segmentation. We demonstrate that our approach is able to distinguish between an independent model and functional copies of the watermarked model with high probability.

Our contributions are summarized as follows:

- We introduce ActiveMark, an approach for watermarking visual foundation models. The method embeds digital watermarks into the hidden representations of a hold-out set of input images; the appropriate representation is selected based on the concept of massive activations.

- Experimental results indicate that the method can differentiate between a watermarked VFM and independently trained models, even after the original model has been fine-tuned for the downstream tasks such as classification and segmentation.

- We theoretically derive an upper bound on the probabilities of false positive detection of a non-watermarked model and misdetection of a functional copy of the watermarked model.

- We demonstrate that the method can be applied to different VFM architectures, suggesting its practical applicability. To our knowledge, this is the first work that addresses the problem of watermarking of visual foundation models.

## 2 RELATED WORK

### 2.1 MASSIVE ACTIVATIONS IN FOUNDATION MODELS

Visual foundation models, particularly those using vision transformers (ViT, Dosovitskiy et al. (2021)), are widely used in modern computer vision due to their scalability and transferability across tasks. The advancement of self-supervised learning methods (Balestriero et al.) has facilitated the creation of general-purpose models, including SimCLR (Chen et al., 2020), DINO (Caron et al.,

2021), CLIP (Radford et al., 2021), and DINOv2 (Oquab et al., 2024). These models learn representations from unlabeled images and demonstrate broad applicability across diverse tasks, often requiring minimal labeled data for fine-tuning. Nevertheless, their internal mechanisms, specifically the characteristics of neural activations, remain an area of ongoing research.

A recent concept in the analysis of these models is massive activations (Sun et al., 2024) – unusually high response values in specific layers or tokens that contribute significantly to the model's output. These activations have been observed to occur across the latter layers of a model, are often of consistently high magnitudes, and can be located at the same spatial or token positions across different input images.

## 2.2 PROTECTING INTELLECTUAL PROPERTY OF NEURAL NETWORKS

The use of watermarking and fingerprinting techniques to protect the intellectual property of neural networks has received increased attention within the field of trustworthy artificial intelligence. Research in this area includes various approaches: for instance, Xu et al. (2024a) applies instruction tuning to fingerprint large language models, using a predefined private key to trigger a specific text output. In Song et al. (2024), the definitions of artifact and fingerprint in large generative models are formalized based on the geometric properties of the training data manifold. Another approach, proposed in Pautov et al. (2024), involves using artificially generated images for the attribution of image classifiers under model extraction attacks. Furthermore, research in Sander et al. (2024) indicates that it is possible to detect if a model was trained on the synthetic output of a watermarked large language model, highlighting a potential privacy consideration associated with neural network watermarking.

## 3 METHODOLOGY

### 3.1 PROBLEM STATEMENT

In this work, we focus on the problem of watermarking of visual foundation models. To describe the proposed method, we start by introducing the notations. Let $s$ be the dimension of the input image, $f : \mathbb{R}^s \to \mathbb{R}^d$ be the source VFM that maps input images to the embeddings of dimension $d$, $\Omega_f$ be the space of visual foundation models that are functionally dependent on $f$, and $\Xi$ be the space of functionally independent visual foundation models. Let $h$ be the dimension of the hidden representation of the image and $m \in \{0, 1\}^n$ be the binary vector of length $n$. In addition, let $f$ be the composition $f(x) \equiv q(p(x))$, where $p : \mathbb{R}^s \to \mathbb{R}^h$ maps an image to the hidden representation, and $q : \mathbb{R}^h \to \mathbb{R}^d$ maps the vector from the hidden representation to the output embedding of the VFM.

In our method, we train two auxiliary models, the encoder $e : \mathbb{R}^h \times \{0, 1\}^n \to \mathbb{R}^h$ that embeds the binary message $m$ into $p(x)$, and the decoder $d : \mathbb{R}^d \to \{0, 1\}^n$ that extracts binary message from the output embedding of the VFM. Additionally, we fine-tune the latter part of the source model, namely, $q$.

Given the image $x$, the message $m$ embedded into $p(x)$ and the parametric transform $\pi(f|\theta) : \Omega_f \to \Omega_f$ that maps the foundation model to its functional copy from $\Omega_f$, the goal of the method is twofold: on the one hand, the decoder $d$ should extract close messages from hidden representations of $f$ and of instances from $\Omega_f$; on the other hand, given the model $g \in \Xi$ which is functionally independent of $f$, the messages extracted from hidden representations of $f$ and $g$ should be far apart.

The formal problem statement goes as follows. Given a predefined threshold $\tau \ll n$ and probability thresholds $\gamma_1 \gg 0, \gamma_2 \ll 1$, the following inequalities should hold:

$$
\begin{cases}
\mathbb{P}_{f' \sim \Omega_f} \left( \|d(q(e(p(x), m))) - d(q(e(p'(x), m)))\|_1 \leq \tau \right) \geq \gamma_1, \\
\\
\mathbb{P}_{g \sim \Xi} \left( \|d(q(e(p(x), m))) - d(q(e(\hat{p}(x), m)))\|_1 \leq \tau \right) \leq \gamma_2,
\end{cases}
\tag{1}
$$

where $f'(x) \equiv q'(p'(x))$ and $g(x) \equiv \hat{q}(\hat{p}(x))$.

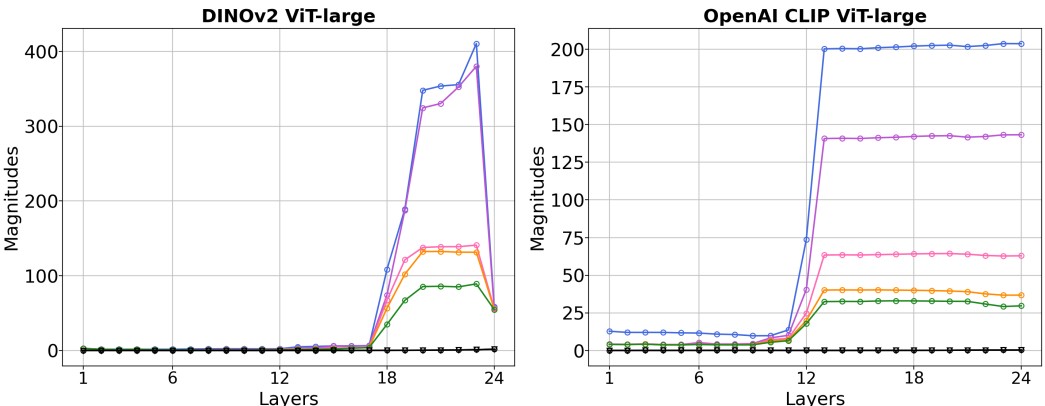

Figure 2: The average magnitudes of activations of blocks of the source VFMs. It is noteworthy that starting from a particular block, the magnitudes of activations increase drastically, namely, starting from block 18 of DINOv2 and from block 12 of CLIP.

### 3.2 ACTIVEMARK IN A NUTSHELL

We introduce ActiveMark, a novel watermarking approach designed specifically for visual foundation models. ActiveMark embeds user-specific binary signatures into a preselected internal feature representation of a holdout set of input images. To do so, we fine-tune a small number of latter layers of the source VFM together with training the lightweight encoder and decoder networks. This approach enables ownership verification by extracting digital fingerprints directly from the model's activations when provided with specific input images. Unlike traditional watermarking techniques that modify model weights or outputs, our method introduces minimal architectural changes while preserving the functional capacity of the model.

The watermarking procedure goes as follows. First of all, given input image $x$ and user-specific binary signature $m$, we train a small encoder network $e$ that injects $m$ into a selected channel of the internal activation of the preselected transformer block of the source model $f$. Then, the modified representation of $x$, namely, $e(p(x), m)$, is propagated through the latter part of $f$, namely, through $q$. At the last block, the decoder network $d$ extracts the binary message $m'$ from the output embedding of $e(p(x), m)$, namely, $m' = d(q(e(p(x), m)))$.

The encoder, decoder, and the latter part of the foundation model are trained jointly to minimize the discrepancy between $m'$ and $m$ while leaving the output embedding intact.

#### 3.2.1 LOSS FUNCTION

The training objective is the combination of two terms: given the input sample $x$, the first one ensures that the feature representations of the watermarked and original models do not deviate much; the second term forces the extracted binary message to be close to the embedded one. Specifically, given $q = q(x)$ as the internal representation of $x$ produced by the source model, and $\tilde{q} = \tilde{q}(x)$ as the representation produced by the watermarked model, the objective function is

$$L(x, d, e, \tilde{q}) = \|q(x) - \tilde{q}(x)\|_2 + \lambda \|m - m'\|_1, \tag{2}$$

where $\lambda > 0$ is a scalar parameter and $m' = d(\tilde{q}(e(p(x), m)))$ is the binary message extracted by the decoder on a particular training iteration. This formulation ensures the successful embedding and extraction of watermarks with little to no impact on the feature representation.

#### 3.2.2 EVALUATING THE EFFICIENCY OF THE METHOD

To evaluate the performance of the proposed method, given the user-specific message $m$ and input image $x$, we compute the distance between the extracted binary message $m'(f, x)$ and $m$. Note that we specifically indicate that the extracted message depends both on the input image and the model from which it is extracted. We measure the distance as the number of bits which differ in $m$ and

$m'(f, x)$:

$$\rho(m, m'(f, x)) = \sum_{i=1}^{n} \mathbb{1}(m_i \neq m'(f, x)_i) \tag{3}$$

Recall that a good watermarking method has to satisfy two conditions: on the one hand, given input image $x$, for the watermarked model $f$, the distance has to be close to 0; on the other hand, for a separate (independent) model, the distance has to be close to $n$. In this work, the decision rule that is used to evaluate whether the given network is watermarked is the comparison of the distance with a predefined threshold: given a single input image $x$, we treat $f$ as watermarked iff

$$\rho(m, m'(f, x)) \leq \tau, \tag{4}$$

where $\tau \geq 0$ is the threshold value. In the case of many input images used for watermarking, namely, for $N$ images from $\mathcal{X} = \{x_1, \ldots, x_N\}$, the performance of the method is illustrated by the watermark detection rate, $R$, in the form below:

$$R = R(f, \mathcal{X}, \tau) = \sum_{i=1}^{N} \mathbb{1}[\rho(m(x_i), m'(f, x_i)) \leq \tau]. \tag{5}$$

### 3.2.3 SETTING THE THRESHOLD VALUE

We set the threshold by formulating a hypothesis test: the null hypothesis, $H_0 = $ "the model $f$ is not watermarked", is tested against an alternative hypothesis, $H_1 = $ "the model $f$ is watermarked", for the given model $f$. In this section, we assume that the probabilities that the $i'$th bit in $m$ and $m'(g, x)$ coincide are the same for all $i \in [1, n]$. Having said so, we estimate the probability of false acceptance of hypothesis $H_1$ (namely, $FPR_1$) as follows:

$$FPR_1 = \mathbb{P}_{g \sim \Xi}[\rho(m, m'(g, x)) < \tau] = \sum_{j=0}^{\tau} \binom{n}{j}(1-r)^j r^{n-j}, \tag{6}$$

where $r = \mathbb{P}_{g \sim \Xi}(m_i = m'(g, x)_i)$. To choose a proper threshold value for $\tau$, we set up an upper bound for $FPR_1$ as $\varepsilon$ and solve for $\tau$, namely,

$$\tau = \arg\max_{\tau' < n} \left( \sum_{j=0}^{\tau'} \binom{n}{j}(1-r)^j r^{n-j} \right) \quad \text{s.t.} \quad \sum_{j=0}^{\tau'} \binom{n}{j}(1-r)^j r^{n-j} < \varepsilon. \tag{7}$$

### 3.2.4 SELECTION OF THE INTERNAL REPRESENTATION

The core idea behind ActiveMark is to embed binary signatures into expressive regions of a model's latent space. We build on the observation that massive activations tend to emerge in later blocks of VFMs and appear for the majority of the input images. Hence, we hypothesize that these high-activation regions are suitable for watermark embedding due to their huge impact on the image representations in the subsequent blocks of the model (Sun et al., 2024).

To identify such regions, we analyze the activation patterns across the blocks of a pre-trained VFM. Specifically, we pass 100 randomly selected natural images through the model and compute, for each block, the average of the top-5 activation magnitudes per image. These per-block averages are then aggregated over all images to yield a global activation profile. As shown in Fig. 2, we observe an explosion of activation magnitudes in the final blocks of the model architecture. We hypothesize that the first block, where massive activations occur, is suitable to embed watermarks into.

To verify this assumption, we study the dependence of the watermark detection rate on the number num of the last frozen block of the source watermarked VFM (see Fig. 1). Specifically, we set two threshold values, $\tau = 0$ and $\tau = 4$, and compute the average watermark detection rates and average bitwise distance between an embedded watermark and an extracted binary message. Note that the value of the watermark detection rate at $\tau = 0$ indicates the ability of the method to extract the same watermarks that were embedded. To evaluate the robustness of the embedded watermarks, we fine-tuned the watermarked CLIP model for the semantic segmentation task. The results are reported in Table 1.

Table 1: Dependency of the watermark detection rate, $R(\tau)$, on the number of the first expressive transformer block, `num`. The architecture of the source VFM is CLIP.

| Block number, `num` | $R(\tau = 0) \uparrow$ | Avg. error $\downarrow$ | $R(\tau = 4)$, finetuned VFM $\uparrow$ |
|---|---|---|---|
| 1 | 0.000 | 15.959 | 0.000 |
| 2 | 0.000 | 16.033 | 0.000 |
| 10 | 0.000 | 14.382 | 0.000 |
| 11 | 0.000 | 11.255 | 0.000 |
| 12 | 0.938 | **0.143** | **0.983** |
| 13 | **0.960** | 0.607 | 0.980 |
| 14 | 0.483 | 1.766 | 0.610 |
| 15 | 0.940 | 0.885 | 0.810 |
| 21 | 0.939 | 0.328 | 0.945 |
| 22 | 0.931 | 0.150 | 0.864 |
| 23 | 0.569 | 0.852 | 0.882 |
| 24 | 0.000 | 15.905 | 0.000 |

The results indicate that early transformer blocks have low watermark detection rates and high reconstruction errors, making them unsuitable for embedding. In contrast, blocks $12, 13, 15, 21$, and $22$ show high detection rates and low errors, demonstrating better stability and reliability for watermark embedding. Block 12, in particular, provides the best balance between detection accuracy and robustness to finetuning. Notably, this is also the first layer where massive activations begin to emerge, which may contribute to its effectiveness as an embedding point. Thus, we decided to use this block as the one to embed watermarks into.

### 3.3 DIFFERENCE BETWEEN WATERMARKED AND NON-WATERMARKED MODELS

Recall that a good watermarking approach should yield a high watermark detection rate from equation 5 for the models that are functionally connected to the watermarked one and, at the same time, low detection rates for independent models. To assess the integrity of the proposed approach, we estimate the probabilities of the method to yield low detection rates for functionally dependent models and high detection rates for independent models in the form

$$\mathbb{P}_{f' \sim \Omega_f}[R(f', \mathcal{X}, \tau) < \overline{R}], \quad \mathbb{P}_{g \sim \Xi}[R(g, \mathcal{X}, \tau) > \underline{R}] \tag{8}$$

for some threshold values $0 < \underline{R} < \overline{R} < N$.

To estimate the probabilities from equation 8, we firstly provide one-sided interval estimations for conditional probabilities of bit collisions in the form

$$r(\Omega_f|x) = \mathbb{P}_{f' \sim \Omega_f}[m_i = m'(f', x)_i], \quad r(\Xi|x) = \mathbb{P}_{g \sim \Xi}[m_i = m'(g, x)_i]. \tag{9}$$

We do it by sampling $M$ functionally dependent models, namely, $f'_1, \ldots, f'_M \sim \Omega_f$, and $M$ independent models, namely $g_1, \ldots, g_M \sim \Xi$. Here, the space $\Xi$ of independent models consists of visual foundation models, both of the same architecture and of different architectures as $f$, by either fine-tuning of *non-watermarked* copy of $f$ for a downstream task, of via functionality stealing perturbations, for example, via knowledge distillation (Hinton, 2014) or pruning (Han et al., 2016). Similarly, the space $\Omega_f$ consists of the models, both of the same architecture and of different architectures as $f$, by either fine-tuning of $f$ for a downstream task, of via functionality stealing perturbations.

Then, given the set $\mathcal{X} = \{x_1, \ldots, x_N\}$ of images used for the watermarking of $f$ from equation 5, we compute the quantities

$$\mathbb{1}(f'_j, i, x_l) = \mathbb{1}[m_i = m'(f'_j, x_l)_i] \quad \text{and} \quad \mathbb{1}(g_j, i, x_l) = \mathbb{1}[m_i = m'(g_j, x_l)_i] \tag{10}$$

as the samples to estimate the probabilities from equation 9 and compute one-sided Clopper-Pearson confidence intervals for $r(\Omega_f|x)$ and $r(\Xi|x)$ in the form

$$\begin{cases} \mathbb{P}(r(\Omega_f|x) < l(x)) \le \frac{\alpha}{N}, \\ \mathbb{P}(r(\Xi|x) > u(x)) \le \frac{\alpha}{N}. \end{cases} \tag{11}$$

These estimates, namely, $l(x)$ and $u(x)$, are used to estimate the probabilities from equation 8.

### 3.3.1 Estimating the probability of a deviation of the detection rate

In this section, we discuss how to upper-bound both the probability of false detection of a non-watermarked model as a copy of the watermarked one and the probability of misdetecting a functional copy of the watermarked model.

Note that $R(f', \mathcal{X}, \tau)$ is a sum of $N$ independent Bernoulli variables with parameters, $r(\Omega_f|x)$, so

$$\mathbb{P}_{f' \sim \Omega_f}[R(f', \mathcal{X}, \tau) < \overline{R}] = \sum_{l=0}^{\overline{R}-1} \sum_{S \subset \mathcal{X}:|S|=l} \prod_{x_{in} \in S} r(\Omega_f|x_{in}) \prod_{x_{out} \notin S} (1 - r(\Omega_f|x_{out})). \quad (12)$$

Note that replacing the parameters $r(\Omega_f|x)$ with its estimations in the form $l(x)$ from equation 11 yields the bound

$$\mathbb{P}_{f' \sim \Omega_f}[R(f', \mathcal{X}, \tau) < \overline{R}] < \sum_{l=0}^{\overline{R}-1} \sum_{S \subset \mathcal{X}:|S|=l} \prod_{x_{in} \in S} l(x_{in}) \prod_{x_{out} \notin S} (1 - l(x_{out})) = \underline{p}(\Omega), \quad (13)$$

that holds with probability at least $1 - \alpha$. Similarly,

$$\mathbb{P}_{g \sim \Xi}[R(g, \mathcal{X}, \tau) > \underline{R}] < \sum_{l=\underline{R}+1}^{N} \sum_{S \subset \mathcal{X}:|S|=l} \prod_{x_{in} \in S} u(x_{in}) \prod_{x_{out} \notin S} (1 - u(x_{out})) = \underline{p}(\Xi). \quad (14)$$

**Remark.** *During experimentally, we used $n = 32, \tau = 5, M = 1000$ and varied confidence level $\alpha$ such that probabilities $\alpha, \underline{p}(\Xi), \underline{p}(\Omega)$ were close. Specifically, value $\alpha = 5 \times 10^{-6}$ yield $\underline{p}(\Omega) = 10^{-6}, \underline{p}(\Xi) = 10^{-4}$ and $\overline{R} = 750, \underline{R} = 600$.*

Thus, if one uses the boundary values $\underline{R}, \overline{R}$ to distinguish between the watermarked and non-watermarked model, one is guaranteed to have both error probabilities $\underline{p}(\Omega), \underline{p}(\Xi)$ low.

## 4 Experiments

We conducted our experiments using two large-scale VFMs, CLIP (Radford et al., 2021) and DINOv2 (Oquab et al., 2024). To train models on downstream tasks (namely, for classification and segmentation), we utilized three domain-specific datasets:

- E-commerce Product Images: This dataset consists of $18,175$ product images categorized into 9 major classes based on Amazon's product taxonomy. It is primarily used for image-based product categorization.

- Oxford-IIIT Pet Dataset: A classification and segmentation dataset containing 37 pet categories (dogs and cats), with approximately 200 images per class. It includes both breed labels and foreground-background segmentation masks.

- FoodSeg103: A food image segmentation dataset containing $7,118$ images annotated with fine-grained pixel-wise labels for over 100 food categories. It supports both semantic segmentation and instance-level analysis of food items.

### 4.1 Watermark injection

Both source VFM were initialized with publicly available pretrained weights. To embed the watermarks, we use a random subset of $N = 1000$ images from the ImageNet dataset Deng et al. (2009). For each image, we randomly and uniformly sample a binary vector $m$ of size $n = 32$ and assign it as the corresponding watermark for this image. Later, we concatenate the first feature channel of $p(x)$ with the message $m$ to obtain the vector $v = \texttt{concat}(p_1(x), m)$ which is passed to the encoder $e$. The output of the encoder, namely, $e(v) \in \mathbb{R}^h$ then replaces the first feature channel of $p(x)$. Next, an updated representation $\tilde{p}(x)$ that differs from the original $p(x)$ in the first channel is passed to the latter part of the VFM, namely, to $q$. The output of the VFM in the form $u = q(\tilde{p}(x))$ is passed to the lightweight decoder, $d$, that produces a vector $d(u)$ of size $n$. An extracted binary message, $m'$, is binarization of $d(u)$ in the form $m'_i = \mathbb{1}(d(u)_i \geq 1/2)$. The schematic illustration of the method is presented in Fig. 1.

## 4.2 FUNCTIONAL PERTURBATIONS OF VFM

To illustrate the robustness of watermarks embedded by ActiveMark, we evaluate how the detection rate from equation 5 changes under finetuning of the model for downstream tasks and under pruning. Specifically, we finetune all the layers of the watermarked VFM for both classification and segmentation downstream tasks, using the aforementioned datasets. For classification tasks, we evaluate three learning rate schedulers: constant (no scheduler), cosine annealing, and linear decay; for the segmentation tasks, no learning rate schedulers were used. The finetuning was performed using the AdamW optimizer for 10 epochs.

To investigate the impact of model sparsity on both classification accuracy and watermark robustness, we applied post-training unstructured $l_1$-norm pruning to the entire model. We evaluated two sparsity levels: moderate pruning, where 20% of the lowest-magnitude weights were zeroed out, and aggressive pruning, where 40% of the weights were removed. This procedure enabled us to assess the effect of varying sparsity levels on watermark reconstruction. Note that the unrestricted $l_1$-norm pruning is used purely as the baseline to illustrate the robustness of the proposed method to the modifications of the model.

## 5 RESULTS

Experimentally, we assess the efficiency of ActiveMark by computing the average watermark detection rate from equation 5 for different values of maximum number of bit errors, $\tau$. In Fig. 3, we demonstrate that our method can be used to reliably detect models that are functionally connected to the watermarked one, namely, the ones obtained via finetuning for downstream tasks and pruning. At the same time, ActiveMark does not falsely detect the presence of the watermarks in negative suspect models.

### 5.1 COMPARISON WITH THE BASELINE WATERMARKING APPROACHES

We indicate the lack of watermarking methods designed specifically for visual foundation models. To compare our approach with some of the general-purpose watermarking approaches, we add a finetuned classification head to the source VFM. The classification head concatenates the CLS token with the mean of patch tokens, applies normalization and dropout, and feeds the result into a linear classifier. Here, the classification backbone is finetuned on the ImageNet dataset. Thus, we compare watermarking approaches in a classification scenario. Specifically, we compare ActiveMark with ADV-TRA (Xu et al., 2024b) and IPGuard (Cao et al., 2021) and report results in Table 2, where we present average watermark detection rates for both positive and negative suspect models. Specifically, negative suspect models here are the ones of different architecture (DINOv2 with registers and CLIP). There, experiments with the positive suspect models correspond to the ones reported in Fig. 3. It is noteworthy that the proposed method outperforms general-purpose watermarking techniques in terms of watermark detection rate, both for positive and negative suspect models.

Table 2: Quantitative comparison with the general-purpose watermarking methods. We report the average watermark detection rate; the architecture of the source VFM is DINOv2.

| Model type | Experiment | ActiveMark | ADV-TRA | IPGuard |
|---|---|---|---|---|
| | clf (const) | 1.000 | 0.703 | 0.200 |
| | clf (linear) | 0.842 | 0.021 | 0.000 |
| Positive suspect ↑ | clf (cosine) | 0.998 | 0.123 | 0.000 |
| | segmantation | 0.990 | 0.000 | 0.000 |
| | pruning (20%) | 1.000 | 1.000 | 0.530 |
| | pruning (40%) | 0.495 | 0.086 | 0.000 |
| Negative suspect ↓ | DINOv2 w/ registers | 0.000 | 0.012 | 0.010 |
| | CLIP | 0.000 | 0.000 | 0.000 |

To assess the computational complexity of the watermarking methods, in Table 3 we present the time in minutes required to embed and extract the watermarks. All the experiments were conducted on a single GPU Nvidia Tesla A100 80GB.

Table 3: Comparison of computation complexity of the watermarking methods. For ActiveMark, we report cumulative results for $N = 1000$ images used for watermarking.

| Method | Watermark embedding | Watermark extraction |
|---|---|---|
| ActiveMark | 34.63 | 4.02 |
| ADV-TRA | 1663.70 | 3.41 |
| IPGuard | 1868.54 | 11.62 |

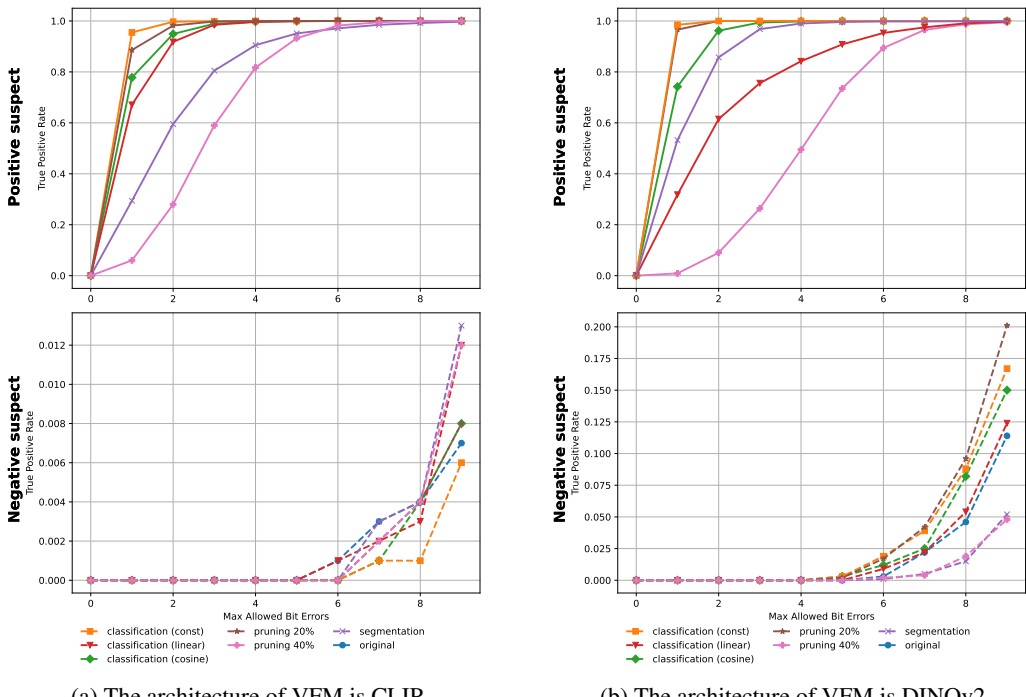

(a) The architecture of VFM is CLIP

(b) The architecture of VFM is DINOv2

Figure 3: Watermark detection rate $R$ from equation 5, averaged over $N = 1000$ images used for watermarking. Classification (const) and classification (cosine) experiments were conducted on the E-commerce Product Images dataset, classification (linear) experiments were conducted on the Oxford-IIIT Pet dataset, and a segmentation experiment was conducted on the FoodSeg103 dataset.

## 6 CONCLUSION

In this work, we propose ActiveMark, a novel watermarking approach for visual foundation models. This method is model agnostic: it is worth mentioning that the model's owner has to prepare a set of input images and perform the watermark embedding procedure only once for a given instance of the model; then, the watermarked model remains detectable by our method after fine-tuning to a particular downstream task (for example, image classification and segmentation). On the other hand, we verify that ActiveMark does not detect benign, independent models as functional copies of the watermarked VFM, which makes the method applicable in practical scenarios. We theoretically show that our method, by design, yields low false positive and false negative detection rates. Possible directions of future work include the adaptation of the proposed approach to a wider range of foundation models.

## 7 REPRODUCIBILITY STATEMENT

In the supplementary material, we provide the source code and configuration for the key experiments, including instructions for embedding and verifying watermarks. In Section 3.3, we provide theoretical justification of the low error probabilities of the proposed method.

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
