# OpenReview forum: "ActiveMark: on watermarking of visual foundation models via massive activations"
_ICLR.cc/2026/Conference — ICLR 2026 Conference Withdrawn Submission_

### Official Review · Reviewer_NqZa · 2025-10-27

**Soundness:** 2
**Presentation:** 1
**Contribution:** 2
**Rating:** 2
**Confidence:** 5

**Summary:**

This paper proposes a watermarking method for Visual Foundation Models (VFMs) to verify ownership. Watermarks are embedded via fine-tuning a small set of layers and a lightweight encoder-decoder network, and remain detectable after downstream fine-tuning. The method is supported by theoretical analysis and experiments, showing low false detection and misdetection rates. However, the experiments are not sufficient.

**Strengths:**

1. Watermarking VFMs is an interesting and promising research direction.

2. The authors upload their source codes, which is good for reproducibility.

3. The paper provides both theoretical analysis and experimental validation, which strengthens the credibility of the approach.

**Weaknesses:**

1. The motivation for using massive activations to embed watermark is unclear. Why is it robust against fine-tuning or pruning attacks? How are channels selected for watermarking? Furthermore, how are blocks chosen for different VFMs, given that identifying blocks with large magnitudes requires manual testing, which limits flexibility. (as illustrated in Figure 2).

2. If the watermarked model is fine-tuned for highly dissimilar downstream tasks, the original watermark can be easily removed, because activations with larger magnitudes, which are crucial for model performance, are more likely to be modified during fine-tuning.

3. This work lacks a clear statement of the Threat Model. The Threat Model should be explicitly formulated, including the adversary’s capabilities, available information, and other aspects, which is essential for any new watermarking approach.

4. The statement of “We indicate the lack of watermarking methods designed specifically for visual foundation models (VFM).” is not incorrect. Various watermarking methods can be applied to VFM, such as weight-based methods [1,2].

[1] Y. Uchida, Y. Nagai, S. Sakazawa, and S. Satoh, “Embedding watermarks into deep neural networks,” in ACM ICMR, 2017, pp. 269–277.

[2] Yao Y, Song J, Jin J. Hashed Watermark as a Filter: Defeating Forging and Overwriting Attacks in Weight-based Neural Network Watermarking[J]. arXiv preprint arXiv:2507.11137, 2025.

5. The related work is not sufficient; there are several activation-based watermark methods [1-3] that have been proposed, but this work neither cites nor compares them.

[1] B. D. Rouhani, H. Chen, and F. Koushanfar, “Deepsigns: an end-to-end watermarking framework for protecting the ownership of deep neural networks,” in ASPLOS, vol. 3, 2019.

[2] Y. Li, L. Abady, H. Wang, and M. Barni, “A feature-map-based large-payload dnn watermarking algorithm,” in IWDW, 2021, pp. 135–148.

[3] J. H. Lim, C. S. Chan, K. W. Ng, L. Fan, and Q. Yang, “Protect, show, attend and tell: Empowering image captioning models with ownership protection,” Pattern Recognition, vol. 122, p. 108285, 2022.

6. In the experiments, this work only considers fine-tuning and pruning attacks, which are relatively simple and insufficient. Other common attacks, such as forging and overwriting, should also be evaluated.

7. This work does not include a fidelity evaluation of the proposed method, which is essential to evaluate its effectiveness.

**Questions:**

Please see the details in the weaknesses.

---

### Official Review · Reviewer_Bp4V · 2025-10-29

**Soundness:** 2
**Presentation:** 2
**Contribution:** 2
**Rating:** 2
**Confidence:** 3

**Summary:**

This paper introduced a ActiveMark watermarking to protect the IP of visual foundation models. Specifically, the model is split into two part frozen layers p() and trainable layers q(). The watermark injection will train three components: via 1) one encoder mapping [m, p(x)] to an vector to replace one channel of the activation p(x), 2) then q() mapping the modified p(x) to the output, and 3) the decoder reconstructing the m from the output. The loss function ensures 1) the outputs between the original q(p(x)) and the q(modified p(x)) as similar as possible, and 2) the reconstructed m’ is similar as m.

**Strengths:**

Strength

1.	Easily implement: It watermarks an already-trained VFM by freezing early layers and training only a small encoder/decoder plus the rear, with a loss that keeps the original representations nearly unchanged.

2.	Model-agnostic: Demonstrated on both CLIP and DINOv2 and presented as applicable across VFM architectures.

**Weaknesses:**

Weakness

1.	Where the watermark is injected? From my perspective, the most important part fronter part of the model is frozen, and the rear part perhaps includes the watermark. In this case, the IP attacker can easily bypass the watermark detection by retraining the rear part, or even directly cutting the rear part. Does the proposed work still work in this case?

2.	White-box requirement: Verification needs access to the suspect model’s internal activation at a specific block to replace one channel with the encoder output before decoding—so pure black-box models (API only) can’t be checked with this method.

3.	Sensitive to the block you choose: If you don’t embed at the “first expressive block” the mark can fail completely; Table 1 shows near-zero detection for early (or too-late) blocks, with strong results mostly around CLIP block 12. This adds a tuning knob and failure mode.

4.	Enough finetuning? The auhtors don’t provide the finetuning details on the VFM. They should use different lr and weight decay and enough epoch to show that their method is robust against the finetuning.

**Questions:**

See my claim in weakness

---

### Official Review · Reviewer_ge8T · 2025-10-30

**Soundness:** 3
**Presentation:** 3
**Contribution:** 3
**Rating:** 4
**Confidence:** 4

**Summary:**

This paper proposes ActiveMark, a watermark for visual foundation models (VFM). The proposed method leverages the massive activation phenomenon in transformer models, where the latter layers of a transformer tend to produce high response values that contribute significantly to the model's final output. ActiveMark embeds watermarks into the layers where mass activation are observed, using a pair of watermark encoder and decoder. Experiment results show that ActiveMark could achieve high watermark detection rate and low false positive rate, as well as robustness against fine-tuning and pruning.

**Strengths:**

+ ActiveMark leverages the massive activation of transformer models to embed watermarks. The watermark design is well-motivated and based on experimental observations and theoretical derivations.
+ ActiveMark could directly be embedded into a pre-trained foundation model without relying on specific downstream tasks. The watermark could remain effective after downstream fine-tuning, making it suitable for protecting general foundation models.

**Weaknesses:**

- **The verification process of ActiveMark requires white-box access to model internals, which might limit its practical use**. Since the watermark is embedded into the intermediate features/layers of the suspect model, one would need access to (part of) the model's internal states and parameters to verify the watermark. This assumption would limit ActiveMark's applicability to black-box scenarios where only the suspect model's input-output pairs are available.
- **The baselines are mainly fingerprinting methods.** The two baselines, ADV-TRA and IPMark are both *fingerprinting* methods, which typically rely on depicting the model's inherent decision boundary and usually do not update the model itself. In contrast, ActiveMark is a *watermarking* method that updates the model to embed watermarks. From this reviewer's perspective, ActiveMark should be evaluated against watermarking methods, rather than fingerprinting ones. While indeed few works exist on directly watermarking transformer-based foundation models, some previous methods have watermarked pre-trained ResNet models [1,2] (which could potentially be applied to transformers as well). Additionally, the authors could consider comparing other watermarking methods using a downstream classification task (as has been done in the current experiment settings).
- **The robustness evaluation does not consider model extraction attacks.** Apart from fine-tuning and pruning, model extraction is also a frequently considered (and usually stronger) watermark removal attack. However, this attack is not evaluated in ActiveMark.

[1] Tianshuo Cong et al., SSLGuard: A Watermarking Scheme for Self-supervised Learning Pre-trained Encoders. CCS '22.
[2] Peizhuo Lv et al., MEA-Defender: A Robust Watermark against Model Extraction Attack. IEEE S&P '24.

**Questions:**

1. Currently the theoretical analysis appears to be a rather general derivation independent of the core method design of ActiveMark. Does the theoretical analysis in Section 3.3 rely on any assumptions specific to ActiveMark? Or does any method that uses multi-bit watermark and hypothesis testing fit into this derivation?
2. How does ActiveMark perform compared to other similar watermarking methods?
3. How does ActiveMark perform against model extraction attacks? Current watermarking methods typically require extra designs for robustness against extraction attacks. It would thus be interesting to see whether ActiveMark (or massive activation in general) naturally transfer to extracted models without relying on additional techniques.
4. How does the watermarked model perform on downstream tasks? Currently the experiment section only reports watermark rate without model performance on the main task.

A few other minor questions:

1. Does massive activation exist in other model architectures (e.g., ResNet)? Could ActiveMark be applied to other architectures than transformers? This is a rather minor question given most foundation models are transformer-based.
2. In Section 4.1, "For each image,we randomly and uniformly sample a binary vector ...": for a single model, does it use a single 32-bit value as its watermark, or is each image assigned a different watermark?
3. Are the results in Table 2 reported on a specific classification/segmentation dataset, or are the results aggregated over all datasets that support classification/segmentation?

---

### Note · Authors · 2025-11-29

I have read and agree with the venue's withdrawal policy on behalf of myself and my co-authors.